# Natural Course of Early Detected Acute Peripancreatic Fluid Collection in Moderately Severe or Severe Acute Pancreatitis

**DOI:** 10.3390/medicina58081131

**Published:** 2022-08-20

**Authors:** Dong Wook Lee, Ho Gak Kim, Chang Min Cho, Min Kyu Jung, Jun Heo, Kwang Bum Cho, Sung Bum Kim, Kook Hyun Kim, Tae Nyeun Kim, Jimin Han, Hyunsoo Kim

**Affiliations:** 1Department of Internal Medicine, School of Medicine, Kyungpook National University, Daegu 41944, Korea; 2Department of Internal Medicine, Daegu Catholic University School of Medicine, Daegu 42472, Korea; 3Department of Internal Medicine, Keimyung University School of Medicine, Daegu 42601, Korea; 4Department of Internal Medicine, Yeungnam University College of Medicine, Daegu 42415, Korea; 5Department of Internal Medicine, Daegu Fatima Hospital, Daegu 41199, Korea

**Keywords:** acute pancreatitis, acute peripancreatic fluid collection, revised Atlanta classification

## Abstract

*Background and Objectives*: Acute peripancreatic fluid collection (APFC) is an acute local complication of acute pancreatitis (AP) according to the revised Atlanta classification. Sometimes APFC resolves completely, sometimes it changes into a pseudocyst or walled-off necrosis (WON), so called late complications. The aim of this study is to investigate the natural course of APFC detected on early computed tomography (CT) in moderately severe (MSAP) or severe AP (SAP). *Materials and Methods*: From October 2014 to September 2015, patients with MSAP or SAP were enrolled if there was APFC within 48 h of onset on imaging studies at six medical centers. The status of fluid collection was followed 4–8 weeks after onset. Initial laboratory findings, CT findings and clinical scoring systems were analyzed. *Results*: A total of 68 patients were enrolled and APFC was completely resolved in 32 (66.7%) patients in the MSAP group and 9 (34.6%) in the SAP group. Patients with a high bedside index for severity in acute pancreatitis (BISAP) score (≥3 points) were common in the SAP group. C-reactive protein (CRP) after 48 h from admission and BUN level were also high in the SAP group. In multivariate analysis, BISAP score (≥3 points), elevation of CRP after 48 h (≥150 mg/L) and nasojejunal feeding after 48 h were risk factors for the development of late complications. *Conclusions*: Spontaneous resolution of APFC was more common in MSAP group and APFC can be changed to pseudocyst or WON in patients with elevated BISAP score, CRP level after 48 h, and non-improved abdominal pain.

## 1. Introduction

Acute pancreatitis (AP) has been reported in 10–50 patients per year for every 100,000 persons and its incidence has been increasing [1]. Furthermore, pancreatic fluid collection has been observed in the surroundings of the pancreas among >50% of patients with AP [2]. In the Atlanta classification for AP published in 1992, pancreatic fluid collection was classified into acute fluid collection, pseudocyst, pancreatic necrosis, and pancreatic abscess [3]. However, opinions on the need for a more specific classification of pancreatic fluid collection have been consistently raised. As a result, the new guideline, named the revised Atlanta classification, was made available in 2012 and has been widely used to date [4].

The revised Atlanta classification states that acute peripancreatic fluid collection (APFC) or acute necrotic collection (ANC) occurs within 4 weeks after AP onset, and late complications such as pseudocyst or walled-off necrosis (WON) may occur after 4 weeks. Among them, APFC is considered one of local complications and if APFC is present in patients with AP, computed tomography (CT) is the best modality to identify APFC as well as other causes of pain; however, CT performed within 48 h can underestimate pancreatic/peripancreatic inflammations and necrosis.

As such, the authors conducted a follow-up investigation for patients who showed APFC on CT performed within the first 48 h among patients admitted with AP to identify the natural course of APFC and the risk factors for the development of late complications.

## 2. Materials and Methods

### 2.1. Study Design

This study prospectively registered 84 patients from six medical centers within Daegu-Gyeongsang province, South Korea, from October 2014 to September 2015. This trial was approved by the institutional review board of each participating medical center and was conducted in accordance with Good Clinical Practices under the principles of the Declaration of Helsinki. This study was funded by Wolbong of the Korean Gastrointestinal Endoscopy Research Foundation.

### 2.2. Inclusion and Exclusion Criteria

The study subjects were recruited from patients with moderately severe (MSAP) or severe AP (SAP) and APFC was shown in CT performed on admission (index CT). The criteria presented in the guidelines were applied for the diagnosis of AP including increasing lipase and/or amylase level [4]. The exclusion criteria were as follows: (1) necrosis observed in either the pancreas or peripancreatic area on index CT; (2) patient with presenting only pancreatic swelling on index CT (without any fluid collection in the peripancreatic area); (3) acute exacerbation of chronic pancreatitis; (4) AP caused by malignancy (such as ampullary cancer or pancreatic cancer), and 5) AP developed after endoscopic retrograde cholangiopancreatography.

### 2.3. Severity Assessment

#### 2.3.1. Based on Clinical Symptoms, Signs and Laboratory Test

Upon admission of the patient to the hospital, the onset of pain was first identified, and the degree of pain was recorded using a visual analogue scale. In addition, erythrocyte sedimentation rate and C-reactive protein (CRP) tests were performed along with complete blood cell count tests, including white blood cell (WBC). Liver function tests, on the basis of blood urea nitrogen (BUN), creatinine level, and arterial blood gas analysis (ABGA) were also performed.

#### 2.3.2. Based on Clinical Scoring System

Any occurrence of organ failure was checked using the modified Marshall score, and the patient was considered to have organ failure if the score was >2 points. Moreover, the patients were checked for systemic inflammatory response syndrome (SIRS) state and bedside index for severity in acute pancreatitis (BISAP) score. The severity of pancreatitis was assessed by measuring the APACHE II and Ranson score on admission and within 48 h after admission.

#### 2.3.3. Based on Initial and Follow-Up CT

CT was performed within 48 h of patient admission and follow-up CT was performed 4–8 weeks later to check if there was either resolution of APFC or onset of complications. Additional CT could be performed according to the researcher’s discretion, even in the case of no clinical improvement during treatment. All CTs were assessed on the basis of the CT severity index introduced by Balthalzar, et al. [5].

### 2.4. Treatment of AP and Its Complications

The patients fasted after admission and total parenteral nutrition was performed, if the patient showed no evidence of bowel movement such as loss of bowel sounds on abdominal auscultation, ileus on X-ray without flatus. Nasogastric tube was not used unless the patient showed paralytic ileus or frequent vomiting. Otherwise, oral intake was continued at the discretion of the researchers, and TPN was stopped if not only oral intake was initiated but also the amount of feeding was sufficient. Enteral feeding was performed via a nasojejunal tube if the patient experienced persistent pain or showed no bowel movement for a long time. It was decided that the diagnosis and treatment processes of AP would be same according to the revised Atlanta classification and recommendations at all participating centers. However, additional CT was performed if the researcher determined that the patient’s condition had deteriorated and additional agents such as antibiotics, protease inhibitors, or octreotide were administered at the discretion of the researchers.

### 2.5. Statistical Analysis

Statistical analysis was performed using SPSS version 19.0 (SPSS Inc., Chicago, IL, USA). Two-sample t test was used for comparison of continuous variables, and χ2 test for categorical variables. A logistic regression was performed to identify the risk factors that hindered APFC resolution, and the odds ratio for each factor was derived.

## 3. Results

### 3.1. Natural Course of APFC

A total of 84 patients were evaluated for APFC during the study period, of which 16 were excluded because no follow-up CT was performed 4–8 weeks later, as a result, 68 patients were enrolled in this study. It was confirmed that 42 (61.8%) of 68 patients presented for MSAP and 26 (38.2%) presented for SAP. Among the 42 patients with MSAP, resolution of APFC was observed in 32 patients (76.2%) but pseudocyst developed in 6 patients (14.3%) and WON in 4 patients (9.5%). On the other hand, 32 patients represented SAP and resolution of APFC was observed in 9 patients (34.6%), pseudocyst in 8 patients (30.8%), WON in 9 patients (34.6%), respectively (Figure 1).

### 3.2. Baseline Characteristics of the Patients and AP

No significant difference was found between APFC with MSAP and APFC with SAP in terms of mean age, sex, height, weight, body mass index, pain onset time (before hospital arrival), and pain score (Table 1). Alcohol intake was found to be the most common cause of AP, followed by gallstones in the MSAP and SAP groups and the frequency of each etiology did not differ significantly.

### 3.3. Laboratory Findings of Initial Status

No significant difference was found between the MSAP and SAP group except for BUN, among the blood tests conducted on admission (Table 2). BUN was significantly higher in the SAP group than in the MSAP group (15.5 mg/dL vs. 26.8 mg/dL, *p* = 0.024). Furthermore, CRP level checked at 48 h after admission was 81.9 mg/L in the MSAP group and 168.7 mg/L in the SAP group, demonstrating significantly different levels (*p* = 0.036). 

### 3.4. Clinical Score and Organ Failure

In MSAP and SAP groups, 2 (4.8%) and 7 (26.9%) patients had a high BISAP score (≥3 points), respectively, and the difference was significant (*p* = 0.019). No significant difference in organ failure was observed between the two groups, including cardiac, respiratory, and renal functions, based on the modified Marshall score (Table 3). Moreover, Ranson score over 3 points occurred in 6 patients (23.1%) in SAP group, while only 5 patients (11.9%) in MSAP group but the difference was not significant (*p* = 0.093). The distribution rate of the patients with APACHE II ≥ 8 points was comparable between the MSAP and SAP groups (14.3% vs. 15.4%, *p* = 0.392).

### 3.5. Nutritional Support

Oral feeding was started in 18 patients (42.9%) and 9 patients (34.6%) at admission in the MSAP and SAP groups, respectively. The rest of the patients were not permitted oral intake due to severe pain or no evidence of bowel movement. When comparing the number of patients who were permitted oral intake after 24 h, 33 patients (78.6%) in the MSAP group and 10 patients (38.5%) in the SAP group started oral feeding, respectively (*p* = 0.029), and only one additional patient was allowed oral intake in the SAP group (9 → 10 patients), on the other hands 15 patients in the MSAP group (18 → 33 patients) after 48 h. Furthermore, enteral feeding via a nasojejunal tube was administered in 1 patient (2.4%) in the MSAP group and 8 patients (30.8%) in the SAP group after 48 h (*p* = 0.006) (Table 4).

### 3.6. Initial Pancreatitis Finding and Appearance of Fluid Collection in CT

In the index CT, there was no difference in multiple fluid collections and a large amount of fluid collection presenting more than 6 cm diameter in the MASP group and SAP groups (Table 5). However, additional CT after 1 week was performed more frequently in the SAP group than in the MSAP group (30.8% vs. 7.1%, *p* = 0.033). 

Late complications such as pseudocyst and WON occurred more frequently in the SAP group (65.4%) compared with the MSAP group (23.8%), and the difference was significant (*p* = 0.029). Concerning late complications, proportion of pseudocyst development did not differ between the two groups (*p* = 0.119) but WON was observed in 9 patients (34.6%) in the SAP group and only 4 patients (9.5%) in the MSAP group (*p* = 0.018).

In the MSAP group, additional treatment was necessary in 2 patients for the treatment of late complications, and each was treated with percutaneous and endoscopic drainage. However, percutaneous procedure was performed in 2 patients (7.7%) and endoscopic procedure in 4 patients (15.4%). Surgical treatment was performed in 1 patient with SAP and none in the MSAP.

### 3.7. Risk Factors for Late Complications after APFC

If the initial severity of AP was severe AP, the APFC tended to change into late complications such as pseudocyst or WON and CRP level more than 150 mg/L at 48 h after admission also were confirmed to cause late complications. Moreover, BISAP score more than 3 points at the time of admission showed an increased risk of late complications. Late complications tended to occur in cases of enteral feeding via a nasojejunal tube or additional CT after 48 h from admission. However, the increased BUN level (≥20 mg/dL) at the time of admission was not a risk factor according to the multivariate analysis (Table 6).

## 4. Discussion

This study examined the proportion of patients who showed late complications among the enrolled subject and the potential risk factors that are associated with late complications when performing follow-up CT in patients of APFC with MSAP and SAP. The results of the investigation confirmed that late complications such as pseudocyst or WON occurred in 23.8% and 65.4% of patients in the MSAP and SAP groups, respectively, and WON was more common in the SAP group. The potential risk for late complications was higher in patients with CRP level ≥ 150 mg/L and BUN ≥ 20 mg/dL after 48 h from admission. Moreover, nasojejunal feeding started after 48 h from admission also has a high potential for late complications, and it was more frequent to perform additional CT after 48 h in patients with late complications.

In a study of 5,819 patients from 69 institutions, mortality increased when BUN increased by more than 5 mg/dL within 24 h of hospitalization for AP [6]. Another study reported that mortality increased when the serum BUN level was greater than 20 mg/dL at the time of hospitalization [7]. BUN reflects the initial condition of the patient and is a useful indicator of the appropriateness of initial resuscitation; however, few studies have reported a positive correlation between BUN and late complications. In this study, BUN levels were significantly higher in the SAP group; however, elevated BUN level was not a risk factor for late complications.

CRP is an acute-phase reactant synthesized in the liver after stimulation by serum interleukin (IL)-1 and IL-6. CRP is a single indicator, commonly used to evaluate the severity of AP. CRP measured 48 h after admission can predict the outcomes of AP [8], and a CRP level more than 150 mg/L at 48 h after admission can predict a worse prognosis of AP in most guidelines on AP [9]. In this study, CRP was a significant risk factor for late complications if it was more than 150 mg/L at 48 h after admission, but the initial CRP level did not differ between the MSAP and SAP groups.

BISAP score predicts severity based on five factors measured 24 h after hospitalization. When no factor is satisfied, mortality is less than 1%; however, mortality increases beyond 22% when all five factors are satisfied [10]. A prospective study showed that the risk of organ failure and pancreatic necrosis significantly increased to 7.4 (95% CI 2.8–19.5) and 3.8 (95% CI, 1.8–8.5), respectively, when the BISAP score was 3 points or higher [11]. In this study, late complications such as pseudocyst or WON developed more frequently in patients with higher BISAP score (≥3 points) presenting with an odds ratio of 2.53.

Continued organ failure for >48 h in patients with AP is classified as SAP, which shows high mortality and complication rates [4,12,13]. The main indication for additional treatment for late complications is clinical suspicion or documented infection of the lesion. Furthermore, patients with significant clinical symptoms arising from pseudocyst or WON requires interventional treatment [14]. The enrolled patients in this study also received interventional or surgical treatment for late complications in cases of infection or symptoms. 

In patients with AP, early initiation of enteral feeding is recommended if the pain is improved and no ileus is observed [15,16]. In this study, oral feeding was also initiated if the pain improved, and any evidence of bowel movement was observed. Patients with late complications complained of continued pain and ileus, which indirectly suggests that inflammation caused by AP is still actively in progress in both the peripancreatic area and abdominal cavity. Moreover, such a long-term inflammatory reaction appears to play a core role in the development of late complications.

The revised Atlanta classification specifies that the degree of local complication cannot reflect the severity of AP, and that the extent of morphological change is not proportional to the severity of organ failure [4]. In this study, the amount or multiplicity of fluid collection on initial CT was not a risk factor for late complications. While APFC appears to be a hypo-attenuated non-enhancing type of fluid on CT, ANC is differentiated in that necrotic tissue is observed as solid rather than fluid [17]. However, a differentiation between APFC and ANC by CT is known to be difficult during the first week after AP onset [18]. Therefore, even if this study also excluded patients who were observed to have definite necrosis in the pancreas or peripancreatic area on CT, it is possible that patients with both pure fluid and necrotic collections were registered in this study and there were more patients in the SAP group who underwent additional CT due to clinical deterioration.

This study had several limitations. First, although the researchers made a consensus on the overall treatment protocol at the multicenter, the guidelines for using protease inhibitors, antibiotics, or anti-secretory drugs were not discussed in advance. Many guidelines still have different levels of recommendations, and no agreement has been made on this yet; thus, further study is required once a specific guideline is presented [15,19,20]. Second, although maximal effort was made to exclude patients with ANC, some patients were expected to be enrolled. However, this study holds significance because APFC in early CT can predict the development of late complications through the initial severity of AP, CRP after 48 h and change in the patient’s condition such as bowel movement.

## 5. Conclusions

APFC in SAP has even more high potential for late complications if it is associated with a high BISAP score (≥3 points) and high CRP level (≥150 mg/L) after 48 h from admission or continued presentation of clinical symptoms reflecting a prolonged inflammatory reaction. We recommend closely observing patient status if APFC is identified on an early imaging study to detect the development of late complications.

## Figures and Tables

**Figure 1 medicina-58-01131-f001:**
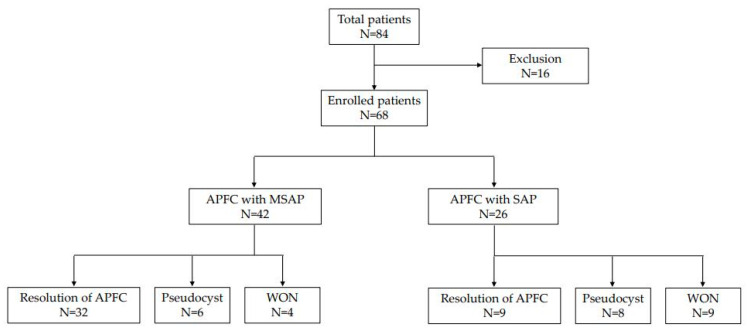
Natural course of peripancreatic fluid collection in all enrolled patients (APFC, acute peripancreatic fluid collection; MSAP, moderately severe acute pancreatitis; SAP, severe acute pancreatitis; WON, walled off necrosis).

**Table 1 medicina-58-01131-t001:** Baseline characteristics and etiology in APFC with MSAP and SAP.

	APFC with MSAP (N = 42)	APFC with SAP (N = 26)	*p*-Value
**Baseline** **characteristics**			
Age, years	52.3 (17.6)	51.6 (13.2)	0.314
Sex, male	28 (66.7)	18 (69.2)	0.683
Height, cm	167.3 (9.6)	166.2 (8.7)	0.516
Weight, kg	68.7 (12.3)	67.1 (13.7)	0.411
BMI, kg/cm^2^	25.8 (3.1)	24.4 (3.9)	0.105
Pain onset time, hours	32.1 (42.6)	33.0 (39.2)	0.797
Pain score, VAS	5.1 (2.1)	5.8 (2.2)	0.197
**Etiology**			
Alcohol	20 (47.6)	13 (50.0)	0.619
Gallstone	11 (26.2)	6 (23.1)	0.785
HyperTG	4 (9.5)	2 (7.7)	0.299
Autoimmune	3 (7.1)	2 (7.7)	0.817
Idiopathic	2 (4.8)	2 (7.7)	0.376
Divisum	2 (4.8)	1 (3.8)	0.772

APFC, acute peripancreatic fluid collection; MSAP, moderately severe acute pancreatitis; SAP, severe acute pancreatitis; BMI, body mass index; VAS, visual analogue scale; hyperTG, hypertriglyceridemia. Values are presented as numbers (%).

**Table 2 medicina-58-01131-t002:** Laboratory findings at admission in APFC patients with MSAP and SAP.

	APFC with MSAP (N = 42)	APFC with SAP (N = 26)	*p*-Value
Hemoglobin, g/dL	13.8 (2.4)	13.2 (2.7)	0.486
Hematocrit	41.1 (6.6)	40.2 (5.1)	0.817
CRP, at admission, mg/L	47.8 (65.8)	59.1 (48.3)	0.284
CRP, after 48 h, mg/L	81.9 (77.3)	168.7 (82.9)	0.036
BUN, mg/dL	15.5 (10.9)	26.8 (11.4)	0.024
Creatinine, mg/dL	0.9 (0.6)	1.1 (0.5)	0.083
Amylase, U/L	378.1 (222.6)	420.9 (312.1)	0.758
Lipase, U/L	614.8 (273.2)	679.6 (477.2)	0.647
pH	7.41 (0.05)	7.40 (0.06)	0.599
pO_2_, mmHg	83.8 (16.1)	82.1 (17.9)	0.231
HCO_3_, mmol/L	22.2 (3.3)	21.3 (4.2)	0.196

APFC, acute peripancreatic fluid collection; MSAP, moderately severe acute pancreatitis; SAP, severe acute pancreatitis; CRP, C-reactive protein; BUN, blood urea nitrogen. Values are presented as mean (standard deviation).

**Table 3 medicina-58-01131-t003:** Clinical scales and organ failure in APFC with MSAP and SAP.

	APFC with MSAP (N = 42)	APFC with SAP (N = 26)	*p*-Value
BISAP score ≥ 3	2 (4.8)	7 (26.9)	0.019
Modified Marshall score			
MMS_cardiac ≥ 2	3 (7.1)	3 (11.5)	0.471
MMS_respiratory ≥ 2	4 (9.5)	2 (7.7)	0.865
MMS_renal ≥ 2	4 (9.5)	3 (11.5)	0.755
Ranson score ≥ 3	5 (11.9)	6 (23.1)	0.093
APACHE II ≥ 8	6 (14.3)	4 (15.4)	0.392

APFC, acute peripancreatic fluid collection; MSAP, moderately severe acute pancreatitis; SAP, severe acute pancreatitis; BISAP, bedside index of severity in acute pancreatitis; MMS, modified Marshall score; APACHE, Acute Physiology and Chronic Health Examination. Values are presented as numbers (%).

**Table 4 medicina-58-01131-t004:** Nutritional support in APFC with MSAP and SAP.

	APFC with MSAP (N = 42)	APFC with SAP (N = 26)	*p*-Value
**Feeding at admission**			
NPO	24 (57.1)	17 (65.4)	0.176
Oral feeding	18 (42.9)	9 (34.6)	0.264
**Feeding after 24 h**			
NPO	8 (19.0)	10 (38.5)	0.132
Oral feeding	33 (78.6)	10 (38.5)	0.029
Nasojejunal feeding	1 (2.4)	6 (23.1)	0.015
**Feeding after 48 h**			
NPO	0 (0.0)	0 (0.0)	-
Oral feeding	41 (97.6)	18 (69.2)	0.042
Nasojejunal feeding	1 (2.4)	8 (30.8)	0.006

APFC, acute peripancreatic fluid collection; MSAP, moderately severe acute pancreatitis; SAP, severe acute pancreatitis; NPO, nothing per oral. Values are presented as numbers (%).

**Table 5 medicina-58-01131-t005:** CT findings and additional treatment of APFC with MSAP and SAP group.

	APFC with MSAP (N = 42)	APFC with SAP (N = 26)	*p*-Value
Fluid collections, multiple	19 (45.2)	12 (46.2)	0.729
Fluid collection diameter > 6 cm	12 (26.1)	8 (30.8)	0.136
Additional CT after 1 week	3 (7.1)	8 (30.8)	0.033
Late complications	10 (23.8)	17 (65.4)	0.029
Pseudocyst	6 (14.3)	8 (30.8)	0.119
WON	4 (9.5)	9 (34.6)	0.018
Additional treatment for pseudocyst/WON			
Percutaneous procedure	1 (10.0)	2 (7.7)	0.815
Endoscopic procedure *	1 (10.0)	4 (15.4)	0.766
Surgery	0 (0.0)	1 (3.8)	0.912

APFC, acute peripancreatic fluid collection; MSAP, moderately severe acute pancreatitis; SAP, severe acute pancreatitis; CT, computed tomography; WON, walled off necrosis; * Endoscopic procedure such as endoscopic ultrasonography-guided cystogastrostomy, endoscopic necrosectomy. Values are presented as numbers (%).

**Table 6 medicina-58-01131-t006:** Multivariate analysis of risk factors about late complication.

Variables	Odds Ratio	95% CI	*p*-Value
APFC with SAP	4.02	2.912–28.754	0.015
CRP, after 48 h ≥ 150 mg/L	3.51	1.914–23.988	0.033
BUN ≥ 20 mg/dL	4.11	0.907–13.101	0.108
BISAP score ≥ 3	2.53	1.014–12.487	0.042
Nasojejunal feeding after 48 h (yes)	7.43	6.366–32.474	0.020
Additional CT after 48 h (yes)	5.36	0.816–21.948	0.336

APFC, acute peripancreatic fluid collection; SAP, severe acute pancreatitis; CRP, C-reactive protein; BUN, blood urea nitrogen; BISAP, bedside index of severity in acute pancreatitis; CT, computed tomography; CI, confidence interval.

## Data Availability

The data presented in this study are available on request from the authors.

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
