# Peer review of "Natural Course of Early Detected Acute Peripancreatic Fluid Collection in Moderately Severe or Severe Acute Pancreatitis"

_medicina, 2022, doi:10.3390/medicina58081131_

Round 1

Reviewer 1 Report

The intentions were good, the work was intense, but I read a paper without a proper acute pancreatitis classification: mild, moderate severe, and severe.

By the exclusion criteria point 1) cases of necrosis observed in either the pancreas or peripancreatic area, you excluded or tried to exclude the more severe cases, but as stated in Atlanta classification early CECT does not provide a reliable assessment of pancreatic or peripancreatic necrosis, this is the reason why in the end you included all types of severity in your study, comparing different forms with different outcomes and concluding only about the acute collections!

Due to these flaws, I think your paper should be rewritten and you should start with the beginning – comparing mild cases with APFC with mild cases, severe cases with APFC with severe cases.

As you stated in the abstract: In multivariate analysis, alcohol etiology, received prolonged total parenteral nutrition and nasojejunal feeding after 48 hours were risk factors to develop late complications. Conclusions: 39.7% of APFC were changed to pseudocyst or WON and alcohol etiology, impossibleoral feeding after 48 hours, high Ranson score were risk factors for late complication.

Except for alcohol the others risk factors describe more severe forms according to Atlanta classification 2012. 

Author Response

The intentions were good, the work was intense, but I read a paper without a proper acute pancreatitis classification: mild, moderate severe, and severe.

By the exclusion criteria point 1) cases of necrosis observed in either the pancreas or peripancreatic area, you excluded or tried to exclude the more severe cases, but as stated in Atlanta classification early CECT does not provide a reliable assessment of pancreatic or peripancreatic necrosis, this is the reason why in the end you included all types of severity in your study, comparing different forms with different outcomes and concluding only about the acute collections!

  • We revised the manuscript title “Natural Course of Early Detected Acute Peripancreatic Fluid Collection in Moderately Severe or Severe Acute Pancreatitis“ and classification enrolled patients and data analysis was performed again according to title.

Due to these flaws, I think your paper should be rewritten and you should start with the beginning – comparing mild cases with APFC with mild cases, severe cases with APFC with severe cases.

  • We absolutely agreed your opinion and manuscript was re-written and data analysis was performed again.

As you stated in the abstract: In multivariate analysis, alcohol etiology, received prolonged total parenteral nutrition and nasojejunal feeding after 48 hours were risk factors to develop late complications. Conclusions: 39.7% of APFC were changed to pseudocyst or WON and alcohol etiology, impossibleoral feeding after 48 hours, high Ranson score were risk factors for late complication.

 Except for alcohol the others risk factors describe more severe forms according to Atlanta classification 2012.

  • After new data analysis, alcohol was not risk factor as your comment, so we removed that result.

Reviewer 2 Report

Introduction:

1.       Lines 47 and 48 – please comment on this statement, there is sufficient information in literature regarding this.

Materials and methods:

1.       It is better to speak about patients rather than cases, this relates to whole manuscript

2.       As 8 clinical centres were involved, please comment and explain on diagnostic and treatment algorithms – were all centres following same criteria for diagnostics and treatment protocols. Please add this to paper.

3.       Increase of Lipase and/or amylase levels is one of the diagnostic criteria for pancreatitis, please add this.

4.       What were the indications for antibacterial treatment (which agents were used) and usage of protease inhibitors and octreotide, please comment on this, because these agents are not routinely used especially in oedematous pancreatitis.

5.       All patients were fasting after admission and majority received TPN even after 48 hours, this is not the routine approach especially in oedematous and mild/moderate pancreatitis, please comment on this. Late provision of enteral nutrition can lead to complications.

6.       What were the indications for CT scan within 48 hours from admission and how small/moderate and large collections are defined, please add.

Results:

1.       As pancreatitis was classified according to Atlanta 2012, please add how many patients were suffering mild/moderate/severe pancreatitis in each of the groups. Please focus on this throughout the manuscript otherwise groups are not comparable.

2.       When was CRP levels tested, in table 2 we can see rather low levels of CRP that usually do not correspond to severe forms of the disease and complication rate. What were the maximum CRP levels in both groups, if this is possible please add.

3.       What hydration protocols were used and for how long time especially in mild/moderate and severe pancreatitis groups. What was the daily fluid balance? There is evidence that overhydration can lead to systemic and local complications.

4.       In late complication group endoscopic procedures were performed, please add which procedures.

5.       Was there any correlation observed with the size of the peripancreatic fluid collections and late complications? Was percutaneous drainage somehow associated with the development of late complications and was drained fluid checked for lipase levels, please add and comment.

Discussion:

Please do not repeat results but rather focus on comparing results with internationally available data.

Author Response

Dear reviewer.

Thank you for your nice comments and I learned a lot through your comments.

  1. Lines 47 and 48 – please comment on this statement, there is sufficient information in literature regarding this.

--> As your comment, there are enough studies, so we removed that sentence.

Materials and methods:

  1. It is better to speak about patients rather than cases, this relates to whole manuscript

--> We removed cases and revised to patients

  1. As 8 clinical centres were involved, please comment and explain on diagnostic and treatment algorithms – were all centres following same criteria for diagnostics and treatment protocols. Please add this to paper.

-->  We added “It was decided that diagnosis and treatment process of AP would be same according to revised Atlanta classification and recommendations at all participated centers. However, the additional CT was performed if researcher determine that the patient`s condition is under deterioration and additional agents such as antibiotics, protease inhibitor, or octreotide was administered at the discretion of researchers.” In lines 102 ~ 107.

  1. Increase of Lipase and/or amylase levels is one of the diagnostic criteria for pancreatitis, please add this.

-->We added the sentence “for diagnosis of AP including increasing of lipase and/or amylase level” in lines66-67

  1. What were the indications for antibacterial treatment (which agents were used) and usage of protease inhibitors and octreotide, please comment on this, because these agents are not routinely used especially in oedematous pancreatitis.

-->We added the sentence “additional agents such as antibiotics, protease inhibitor, or octreotide was administered at the discretion of researchers” in lines 105-107

  1. All patients were fasting after admission and majority received TPN even after 48 hours, this is not the routine approach especially in oedematous and mild/moderate pancreatitis, please comment on this. Late provision of enteral nutrition can lead to complications.

-->We absolutely agree for your opinion and did not perform routine fasting in all patients. We added the sentence “The patients fasted after admission and total parenteral nutrition was performed, if the patient showed no evidence of bowel movement such as loss of bowel sounds on abdominal auscultation, ileus on X-ray without flatus” in lines 95-97.

  1. What were the indications for CT scan within 48 hours from admission and how small/moderate and large collections are defined, please add.

--> As you know, acute pancreatitis should be needed to take a differential diagnosis with various disease so early CT scan is helpful for this aspect but there may be underestimate the status of necrosis in pancreatic/peripancreatic area, you mentioned. Aim in this study is to investigate of natural course of early fluid collection in moderate severe or severe acute pancreatitis so we performed CT on admission. So we added the sentence “The study subjects were recruited from patients with moderately severe (MSAP) or severe AP (SAP) and APFC showed in CT performed on admission (index CT)” and revised the amount of fluid collection from small/moderate/large collection to diameter > 6cm. We revised the table 5.

Results:

  1. As pancreatitis was classified according to Atlanta 2012, please add how many patients were suffering mild/moderate/severe pancreatitis in each of the groups. Please focus on this throughout the manuscript otherwise groups are not comparable.

--> We re-wrote the manuscript after making new classification of enrolled patients such as acute peripancreatic fluid collection (APFC) with moderately severe acute pancreatitis (MSAP) and APFC with SAP.

  1. When was CRP levels tested, in table 2 we can see rather low levels of CRP that usually do not correspond to severe forms of the disease and complication rate. What were the maximum CRP levels in both groups, if this is possible please add.

-->  Sorry for mistake. We revised the unit (mg/dL à mg/L).

  1. What hydration protocols were used and for how long time especially in mild/moderate and severe pancreatitis groups. What was the daily fluid balance? There is evidence that overhydration can lead to systemic and local complications.

--> We performed hydration to patients with AP according to Atlanta classification and their recommendations. But the hydration did not differ in both groups and the table about hydration seems to be complex so we removed the table about hydration.

  1. In late complication group endoscopic procedures were performed, please add which procedures.

--> We added footnote in table 5 as you mentioned.

  1. Was there any correlation observed with the size of the peripancreatic fluid collections and late complications?

--> No there was no correlation with size of aPFC and late complications. We added sentence “In index CT, there was no difference in multiple fluid collections and large amount of fluid collection presenting more than 6cm diameter in MASP group and SAP group” in lines 196-197.

Was percutaneous drainage somehow associated with the development of late complications and was drained fluid checked for lipase levels, please add and comment.

--> No there was no association with percutaneous drainage and late complications. Moreover, we did not check drained fluid analysis and added that comments in limitations part.

Discussion:

Please do not repeat results but rather focus on comparing results with internationally available data.

  • We minimize repeat our result and tried to compare results with references.

Round 2

Reviewer 2 Report

No further comments.

Author Response

Thank you for your review.